# Detection of aberrant splicing events in RNA-seq data using FRASER

Christian Mertes [1,6], Ines F. Scheller [1,2,6], Vicente A. Yépez [1,3], Muhammed H. Çelik[1], Yingjiqiong Liang[1], Laura S. Kremer[4,5], Mirjana Gusic [4,5], Holger Prokisch [4,5] & Julien Gagneur [1,2,5✉]

Aberrant splicing is a major cause of rare diseases. However, its prediction from genome sequence alone remains in most cases inconclusive. Recently, RNA sequencing has proven to be an effective complementary avenue to detect aberrant splicing. Here, we develop FRASER, an algorithm to detect aberrant splicing from RNA sequencing data. Unlike existing methods, FRASER captures not only alternative splicing but also intron retention events. This typically doubles the number of detected aberrant events and identified a pathogenic intron retention in *MCOLN1* causing mucolipidosis. FRASER automatically controls for latent confounders, which are widespread and affect sensitivity substantially. Moreover, FRASER is based on a count distribution and multiple testing correction, thus reducing the number of calls by two orders of magnitude over commonly applied z score cutoffs, with a minor loss of sensitivity. Applying FRASER to rare disease diagnostics is demonstrated by reprioritizing a pathogenic aberrant exon truncation in *TAZ* from a published dataset. FRASER is easy to use and freely available.

[1] Department of Informatics, Technical University of Munich, Garching, Germany. [2] Institute of Computational Biology, Helmholtz Zentrum München, Neuherberg, Germany. [3] Quantitative Biosciences Munich, Gene Center, Ludwig-Maximilians Universität München, Munich, Germany. [4] Institute of Human Genetics, Helmholtz Zentrum München, Neuherberg, Germany. [5] Institute of Human Genetics, Klinikum rechts der Isar, Technical University of Munich, Munich, Germany. [6]These authors contributed equally: Christian Mertes, Ines F. Scheller. ✉email: gagneur@in.tum.de

It is estimated that about 15–30% of the variants causing inherited diseases affect splicing[1–5]. The underlying mechanisms include skipping, truncation, and elongation of exons as well as intron retention[6,7]. Despite advances in the detection of variants affecting splicing by machine learning[8–11], accurate detections remain limited in particular for deep intronic variants[11]. Therefore, genetic diagnosis guidelines require additional functional evidence to classify a variant as pathogenic[12,13]. Furthermore, many variants affecting splicing, especially deep intronic variants, are ignored by most prediction tools[14] or are missed when whole-exome sequencing or panel sequencing technologies are used[15]. To overcome the limitation of genetic variant interpretation, RNA sequencing (RNA-seq) has gained popularity over the last few years[16–19]. RNA-seq allows not only the validation or invalidation of effects on splicing of variants of unknown significance[16] but also allows the detection of de novo aberrant splicing events transcriptome-wide, including the activation of deep intronic cryptic splice sites[16,17,19].

Three distinct methods developed by (1) Cummings et al.[16], (2) Kremer et al.[17], and (3) Frésard et al.[18] have been employed to call aberrant splicing in RNA-seq data for rare disease diagnostics. Moreover, two additional methods to call aberrant splicing were developed in parallel to this study: LeafCutterMD[20] and SPOT[21]. All five methods make use of the so-called RNA-seq split reads, whose ends align to two separated genomic locations of the same chromosome strand and are, therefore, evidence of splicing events. These methods all consider RNA-seq split reads de novo, i.e., beyond annotated splice sites, because the creation of novel splice sites has a strong pathogenic potential by leading to frameshifts, ablation of protein sequences, or creation of non-functional protein sequences. The first method consists of a combination of cutoffs applied to absolute and relative RNA-seq split read counts[16,19]. The limitation of this method is that statistical significance is not assessed. Furthermore, the cutoffs are not data-driven. In particular, it is unclear whether the requirement that an intron occurs in no other sample[16] or in less than

five affected samples[19] would generalize well to larger cohorts than the ones investigated to date. Instead, Kremer et al.[17] tested the significance of differential splicing using LeafCutter[22], a multivariate count fraction model developed for mapping splicing quantitative trait loci. This approach, along with the more recent LeafCutterMD[20] and SPOT[21], which are also multivariate approaches, allowed controlling for false discovery rate (FDR) and are less dependent on cohort size. One limitation, however, was made evident by Frésard et al.[18], who showed that strong covariations of split-read-based splicing metrics are widespread within RNA-seq compendia. The origins of these covariations may include sex, population structure, or technical biases such as batch effects or variable degree of RNA integrity. Not controlling for these latent confounders can substantially affect the sensitivity of the detection of aberrant splicing events. To address this issue, Frésard et al. corrected split-read-based splicing metrics by regressing out principal components. Aberrant splicing events were then identified using a cutoff on the z scores ($|z| \geq 2$) of these corrected splicing metrics. The major drawback of this approach is that an absolute $z$ score cutoff does not guarantee any control for FDR. Moreover, a $z$ score cutoff amounts to a quantile cutoff when assuming that the data distribution is approximately Gaussian. However, Gaussian approximations may be inaccurate when splicing metrics are based on low split read counts, which occurs on splice sites with low coverages and at repressed splice sites.

Here, we address these issues by developing FRASER (Find RAre Splicing Events in RNA-seq), which is an algorithm that provides a count-based statistical test for aberrant splicing detection in RNA-seq samples, while automatically controlling for latent confounders (Fig. 1). Unlike previous methods, FRASER is not limited to alternative splicing, as it also captures intron retention events by considering non-split reads overlapping donor and acceptor splice sites. The parameters are optimized for recalling simulated outliers by training a so-called denoising autoencoder[23]. FRASER shows substantial improvements against

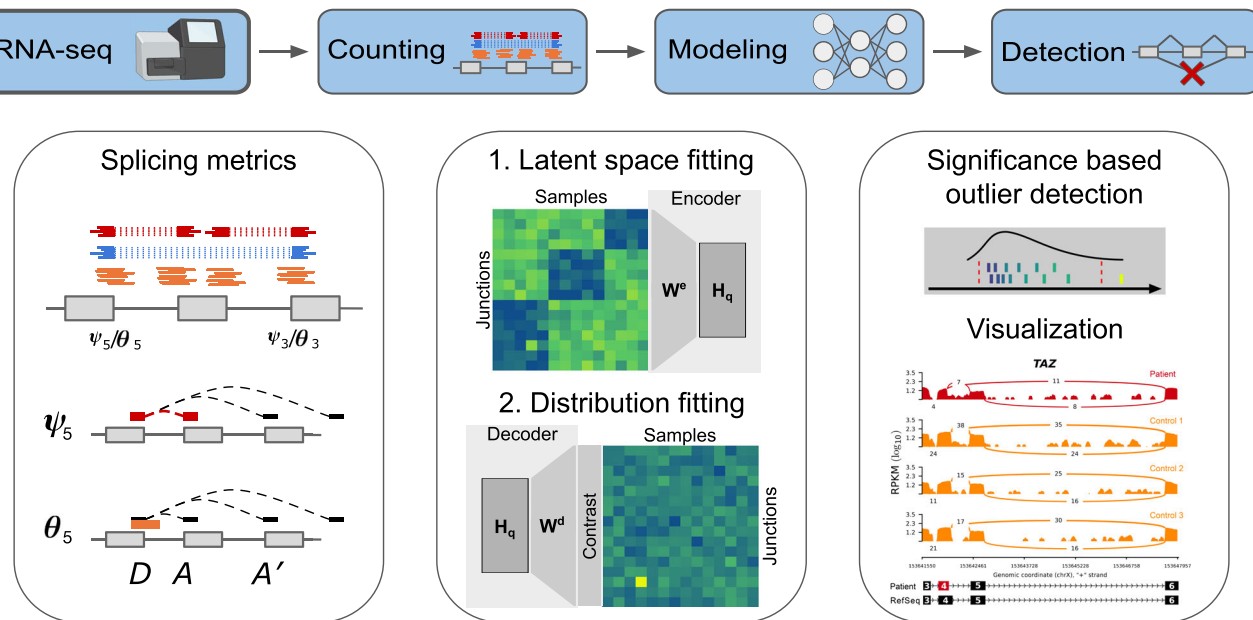

**Fig. 1 The FRASER aberrant splicing detection workflow.** The workflow starts with RNA-seq aligned reads and performs splicing outlier detection in three steps. First (left column), a splice site map is generated in an annotation-free fashion based on RNA-seq split reads. Split reads supporting exon–exon junctions as well as non-split reads overlapping splice sites are counted. Splicing metrics that quantify alternative acceptors ($\psi_5$), alternative donors ($\psi_3$), and splicing efficiencies at donors ($\theta_5$) and acceptors ($\theta_3$) are then computed. Second (middle column), a statistical model is fitted for each splicing metric that controls for sample covariations and overdispersed count ratios. Third (right column), outliers are detected as data points that deviate significantly from the fitted model. Candidates are then visualized using a genome browser. $D$ donor site, $A$ acceptor site. Made in ©BioRender - biorender.com.

former methods on simulations based on the healthy cohort dataset of the Genotype-Tissue Expression project (GTEx)[24]. Lastly, we demonstrate the applicability of FRASER to rare disease diagnosis by reanalyzing an RNA-seq dataset of individuals affected by a rare mitochondrial disorder[17].

## Results

To identify splice sites independently of genome annotation, FRASER creates a splice site map by calling de novo introns supported by a sufficient amount of RNA-seq split reads (Fig. 1, "Methods" section). An intron is defined by a donor (or 5′ splice site) and an acceptor (or 3′ splice site). For each intron, FRASER computes two metrics. The $\psi_5$ metric quantifies alternative acceptor usage. It is defined as the fraction of split reads from an intron of interest over all split reads sharing the same donor as the intron of interest. The $\psi_3$ metric, which is analogously defined for the acceptor, quantifies alternative donor usage. FRASER also considers the donor splicing efficiency metric $\theta_5$, which is defined as the fraction of split reads among split and unsplit reads overlapping a given donor, and the analogously defined acceptor splicing efficiency metric $\theta_3$. Splicing efficiency metrics (denoted collectively $\theta$) have lower values in case of intron retention or impaired splicing. The advantage of these four metrics against alternative splicing metrics, such as the popular percent spliced-in[25], is that they can be quantified from short-read sequencing data without prior exon annotations[26]. These four metrics are read proportions and, therefore, range in the interval [0,1]. For modeling and visualization purposes, we used the corresponding log-odds ratios that were estimated using a robust logit-transformation ("Methods" section).

To establish FRASER, we considered the GTEx project dataset (V6p)[24]. After quality filtering, this dataset consisted of 7,842 RNA-seq samples from 48 tissues of 543 assumed healthy donors. Although the GTEx donors did not suffer from any rare disease, the samples may present aberrant splicing events, just as they present genes with aberrant expression levels[21,27]. After filtering for expressed junctions per tissue ("Methods" section), the FRASER splice site map contained on average 137,058 (±5,848 standard deviation across tissues) donor sites and 136,743 (±5,920) acceptor sites (Supplementary Fig. S1), of which 1.7% and 1.8%, respectively, were not in the GENCODE annotation (release 28)[28]. Hierarchical clustering of intron-centered logit-transformed $\psi_5$ values revealed distinct sample clusters for all GTEx tissues (Fig. 2a–c). Overall, the average absolute correlation between samples per tissue was 0.10 (±0.05 standard deviation across tissues, Fig. 2d). Strong covariation was also observed for $\psi_3$ and for the splicing efficiency metric $\theta$ (Supplementary Figs. S2 and S3). This covariation structure was tissue-specific (Fig. 2a–d). Artefactual covariation due to pseudocount effects could be excluded as sample covariations were even stronger for highly transcribed introns (Supplementary Fig. S4). In some tissues, samples clustered according to the RNA degradation index (e.g., heart left ventricle, Fig. 2b, and Supplementary Figs. S2 and S3b), while in others they clustered according to the sequencing center or to the death classification (e.g., whole blood samples, Fig. 2c). However, no single known covariate could explain covariations for all tissues. Such sample covariations may arise from common genetic variation, technical artifacts, or other unknown factors. These observations, consistent with Frésard et al.[18], motivated us to control for between-sample covariations prior to calling aberrant splicing events.

We modeled those between-sample covariations by fitting a low-dimensional latent space for each tissue separately. The latent space was estimated by principal component analysis (PCA) on logit-transformed splicing metrics. The optimal dimension for the

latent space was determined by maximizing the area under the precision-recall curve when calling artificially injected aberrant values independently for each splicing metric (denoising auto-encoder, "Methods" section). Typically, the value of the latent space dimension $q$ giving the highest area under the precision-recall curve depended on the amplitude of the deviations from the observed values, with higher dimensions (i.e., more complex models) performing better for the milder deviations (Supplementary Fig. S5a). To not depend much on the value of the amplitudes of simulated outliers, we opted for drawing randomly the deviation amplitudes between 0.2 and the maximal possible amplitude a metrics can take (i.e., in order to reach 0 or 1). This strategy is similar to the injection scheme used in OUTRIDER[29]. The method was robust to the choice of the encoding dimension, as the performance for recalling artificial outliers typically plateaued around the optimal dimension (Supplementary Fig. S5a). The fitted encoding dimension per tissue was 15 for $\psi_5$, 16 for $\psi_3$, and 12 for $\theta$ on average. Moreover, the fitted encoding dimension grew approximately linearly with the number of samples resulting in larger encoding dimensions in tissues with more samples (Supplementary Fig. S5b). Controlling for the latent space reduced the between-sample correlation from 0.10 ± 0.05 down to 0.02 ± 0.01 (mean ± standard deviation across tissues; Fig. 2d and Supplementary Figs. S2d and S3d).

**Calling aberrant splicing events using the beta-binomial distribution.** Having established an effective procedure to model between-sample covariations, we then addressed the issue of calling aberrant splicing events by finding statistically significant outlier data points. Based on the latent space, FRASER models the expected value of each observation ("Methods" section). In contrast to methods such as LeafCutter[22], LeafCutterMD[20], and SPOT[21], we modeled each junction individually and did not model jointly all junctions of a gene. We considered the observations that significantly deviated from their expected value as outliers. To this end, we modeled random deviations from the expected values using the beta-binomial (BB) distribution, a distribution for count fractions parameterized by its expected count ratio and an intra-class correlation parameter that accounts for variations exceeding sampling noise ("Methods" section). This model allowed computing a two-sided $P$ value for each observation ("Methods" section). For the alternative acceptor splicing metric $\psi_5$, the $P$ values of introns with the same donor are not independent, as the sum of proportions on which they are based is one. Therefore, we corrected the $\psi_5$ $P$ values for each donor with the family-wise error rate (FWER) using Holm's method, which holds under arbitrary dependence assumption ("Methods" section)[30]. The same approach was applied to $\psi_3$, yielding a single $P$ value per acceptor and sample. Moreover, we controlled the splice site $P$ values for the FDR genome-wide per sample using Benjamini–Yekutieli's method ("Methods" section)[31]. To showcase the application of FRASER, we used the suprapubic skin tissue from the GTEx dataset, as done by Brechtmann et al.[29]. Figure 3a shows as an example the $\psi_5$ metric of the seventh intron of *SRGAP2*, which exhibited a proportional relationship between the number of split reads supporting the seventh intron and the total number of split reads with the same donor site. In this example, the $P$ values tended to be conservative, yet modeling the distribution of the data across samples reasonably well (Fig. 3b). Figure 3c shows an example of an outlier in the $\psi_5$ metric of the 17th intron of *SRRT*, with one data point exhibiting a much higher usage of this acceptor site compared to the other samples and a corresponding very low nominal $P$ value ($P = 5.83 \times 10^{-11}$, Fig. 3d). Across all introns and splice sites, $P$ values were generally conservative (Fig. 3e and Supplementary Fig. S6). An excess

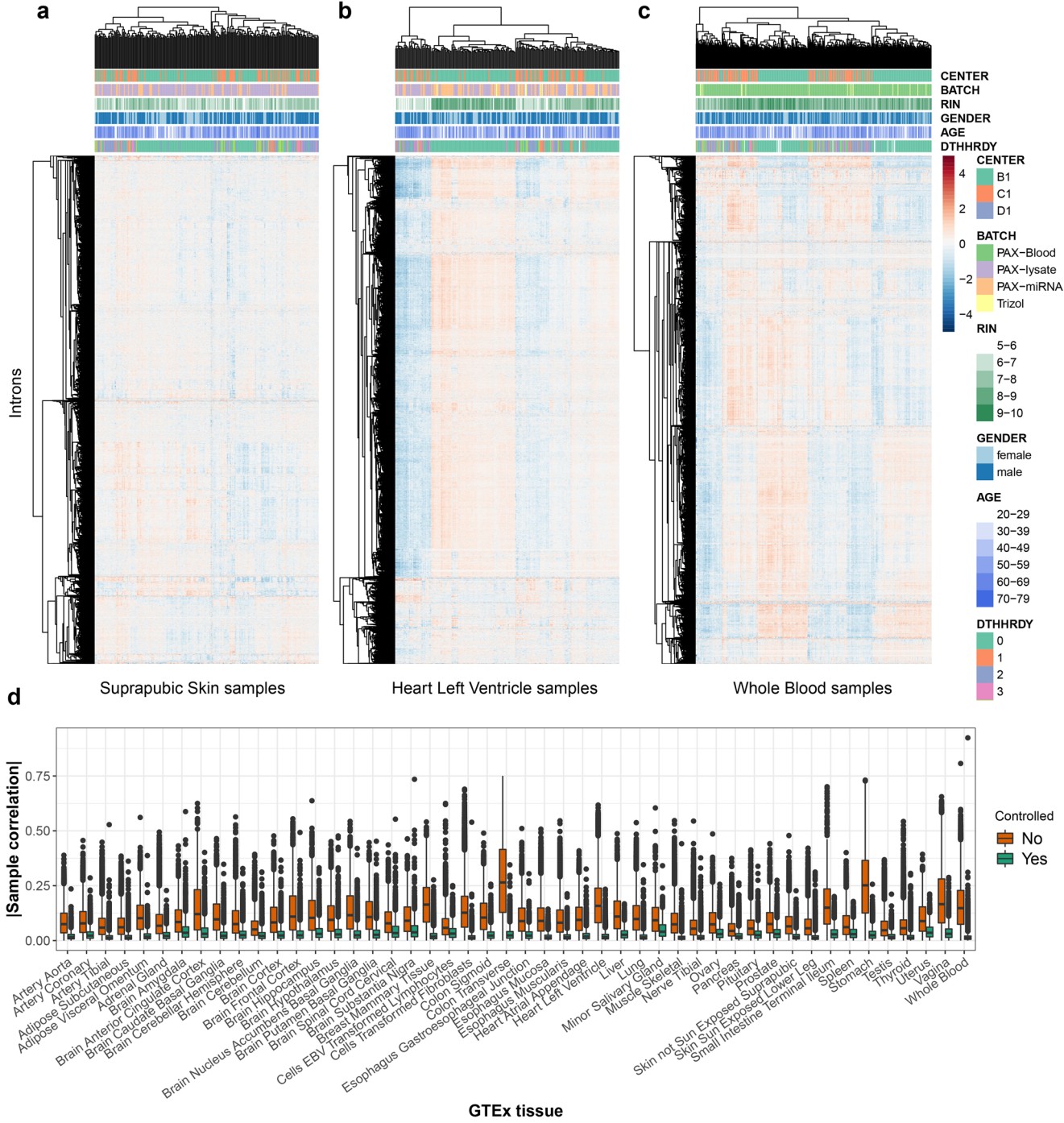

**Fig. 2 FRASER corrects for covariations in alternative acceptor usage. a–c** Intron-centered and logit-transformed $\psi_5$ of the 10,000 most variable introns clustered by samples (columns) and introns (rows) for three representative GTEx tissues: suprapubic skin (**a**, $n = 222$), left ventricle heart (**b**, $n = 211$), and whole blood (**c**, $n = 369$). The red and blue colors indicate relative high and low intron usage, respectively. Colored horizontal tracks display sequencing center, batch, RNA integrity number (RIN), gender, age, and cause of death (DTHHRDY, Hardy scale classification) of the samples. **d** Boxplots of absolute values of between-sample correlations of intron-centered logit-transformed $\psi_5$ for 48 GTEx tissues before (orange) and after (green) correction for the latent space ($n =$ number of sample pairs per GTEx tissue, between 52 and 401 samples per tissue, for more details, see Supplementary Fig. S1E). The $\psi_5$ values were clipped to the [0.01, 0.99] interval before logit-transformation. These plots show that while tissue-specific correlation structures exist among samples, latent space fitting allows correcting for them. The data in **d** are represented as boxplots in which the middle line indicates the median, the bounds of the box indicate the first and third quartiles and the whiskers indicate ±1.5 × IQR (interquartile range) from the third and first quartile, respectively. Outlying data points are shown as dots.

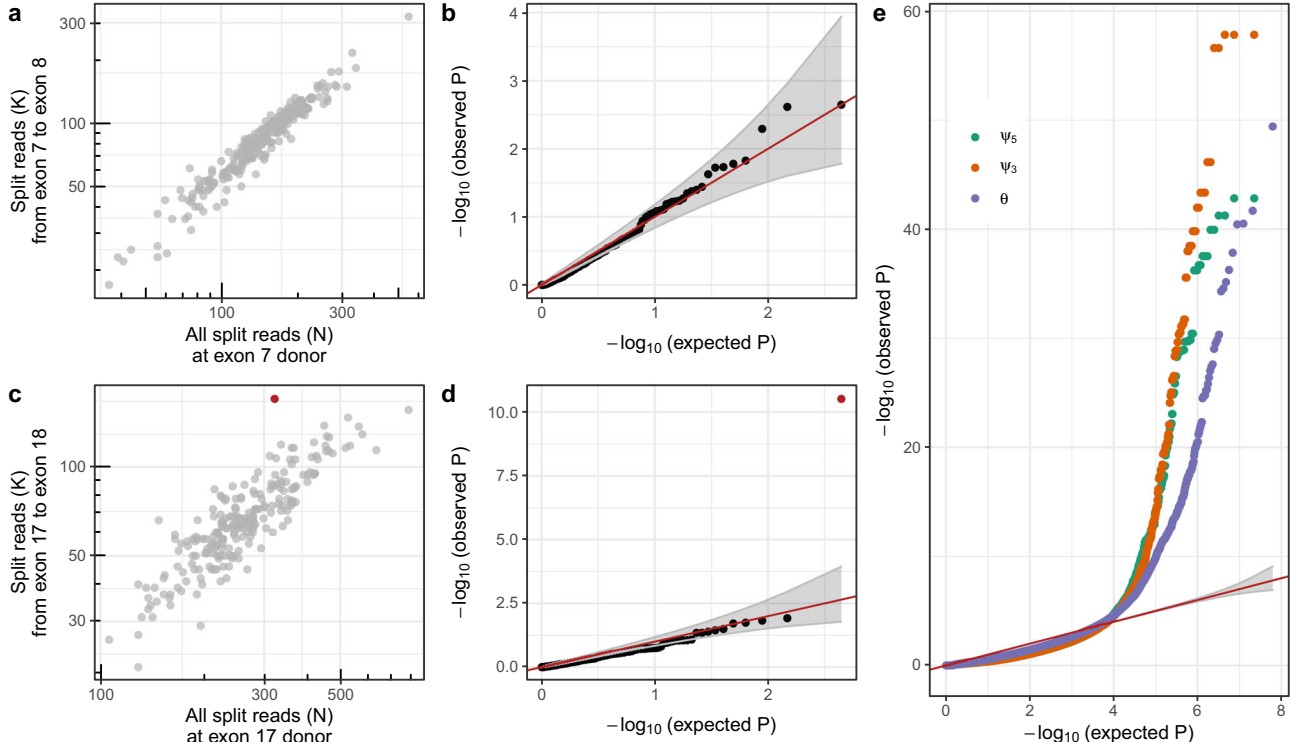

**Fig. 3 Splicing outlier detection based on the beta-binomial distribution. a** Intron split read counts (*y*-axis) against the total donor split read coverage (*x*-axis) for the seventh intron of *SRGAP2*. **b** Observed negative log-transformed *P* values (*y*-axis) against expected ones (*x*-axis) of the $\psi_5$ metric for the data shown in **a**. Under the null hypothesis, the data are expected to lie along the diagonal (red, 95% confidence bands in gray). **c** Same as **a** for the 17th intron of *SRRT*, showing an outlier (FDR < 0.1, red). **d** Same as **b** for the 17th intron of *SRRT*. The outlier is marked in red. **e** Same as **b** across all introns and splice sites for $\psi_5$ (green), $\psi_3$ (orange), and splicing efficiency ($\theta$, purple). **a–e** Based on the suprapubic skin tissue from GTEx (*n* = 222). **b, d, e** *P* values were calculated two-sided with the beta-binomial distribution and significance was determined based on FDR after adjusting for multiple comparisons ("Methods" section). FDR false discovery rate.

of low *P* values was detected for the most extreme ten-thousandth of the data, possibly reflecting genuine aberrant splicing events (Fig. 3e). Similar results were obtained for all GTEx tissues investigated (Supplementary Fig. S7).

**Recall benchmark of artificially injected outliers.** Next, we assessed the performance of FRASER and delineated the contributions of modeling the covariation and of using the BB distribution. To this end, we simulated a ground truth dataset based on the suprapubic skin tissue, in which we artificially injected splicing outliers with a frequency of $10^{-3}$, which yielded 25,988, 26,153, and 49,169 outliers for $\psi_5$, $\psi_3$, and $\theta$, respectively ("Methods" section). The amplitude of the deviations from the original observed values was drawn uniformly between 0.2 and 1 and their directions (increase or decrease) were randomly assigned with equal probability ("Methods" section). We then monitored outlier recall as well as precision, i.e., the proportion of injected outliers among the reported outliers. Methods not modeling covariation performed worse than methods modeling covariation at any level of recall and for all splicing metrics (Fig. 4 and Supplementary Figs. S8–10). Moreover, methods that modeled covariation and used BB-based *P* values yielded a higher precision than those that used *z* scores (Fig. 4 and Supplementary Figs. S8–10). This higher precision was observed at all levels of recalls, simulated outlier amplitudes, and read coverage (Fig. 4). Notably, using PCA and a *z* score cutoff equal to 2, instead of FRASER at FDR < 0.1, yielded two orders of magnitude more outliers across all GTEx tissues (Supplementary Fig. S11) and a drastic drop in precision (3% vs. 92% with FRASER) for a small increase in recall (98% vs. 83% with FRASER, Supplementary

Fig. S12). This drastic difference in precision strongly suggests the advantage of using an FDR cutoff rather than an absolute *z* score cutoff.

The benchmark with simulated outliers also allowed investigating alternative ways to estimate the expected values by regression on the latent space. This included a naïve BB regression, a robust version of the BB regression, as well as a least squares regression of logit-transformed splicing metrics ("Methods" section). The naïve BB regression was too sensitive to outlier data points; hence, it was outperformed by its robust version (Supplementary Fig. S13). However, least squares regression of logit-transformed splicing metrics had a high performance that was similar to that of the robust BB regression (Supplementary Figs. S8–10, "Methods" section) while being much faster to compute. We, therefore, adopted least squares regression of logit-transformed splicing metrics to estimate the expected values.

FRASER considers each junction of a gene individually. In principle, this can be less sensitive than considering all junctions of a gene in a joint model, as with the Dirichlet-Multinomial based methods LeafCutterMD[20], SPOT[21], and the LeafCutter adaptation of Kremer et al.[17]. To assess this, we performed a benchmark whereby splicing outliers are simulated by swapping out, for a single individual, skin and brain tissue read counts for a differentially alternatively spliced gene ("Methods" section). These differentially spliced genes were identified by LeafCutter[22], giving a potential advantage to Dirichlet-Multinomial-based aberrant splicing detection methods. Nevertheless, precision-recall curves for this simulation setting shows that FRASER outperforms the three Dirichlet-Multinomial-based methods (Supplementary

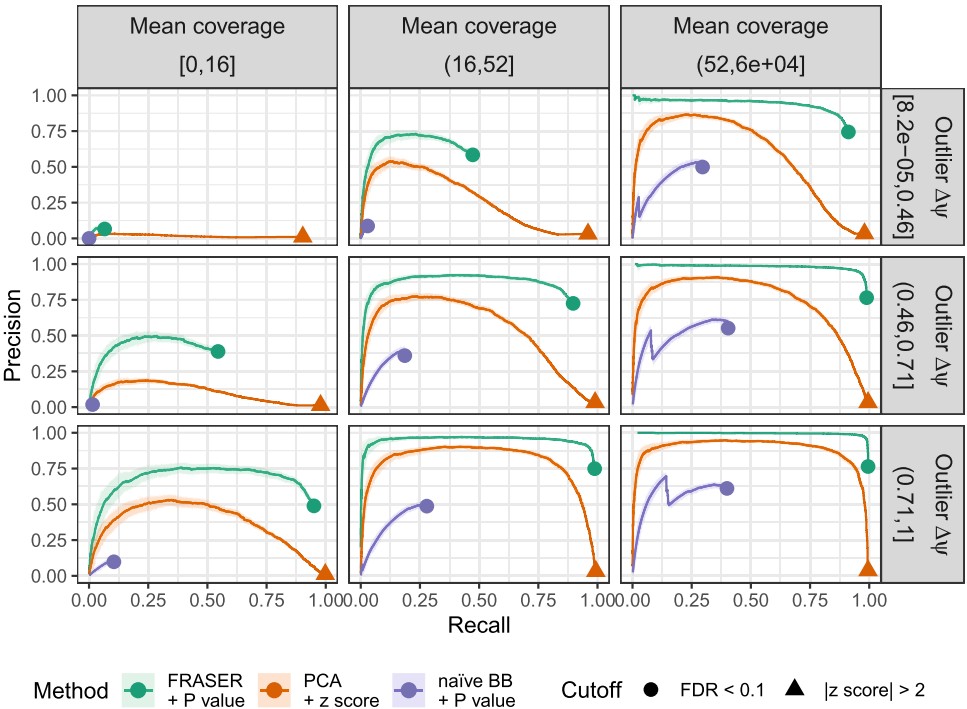

**Fig. 4 Benchmark using artificially injected outliers for alternative acceptor usage.** The proportion of simulated outliers among reported outliers (precision) plotted against the proportion of reported simulated outliers among all simulated outliers (recall) for increasing $P$ values (FRASER, green; naïve beta-binomial regression, purple) or decreasing absolute $z$ scores (PCA, orange). Moreover, all events with $|\Delta\psi_5|<0.1$ are ranked last. The data are stratified by the mean coverage of the intron (columns) and by the injected absolute $\Delta\psi_5$ value (rows). The cutoffs for each method are marked (FDR < 0.1, circle; absolute $z$ score > 2, triangle). The darker lines mark the precision-recall curves computed for the full dataset while the light ribbons around the curves indicate 95% confidence bands estimated by bootstrapping ($n = 200$). These results show the importance of controlling for latent confounders, of using a count-based distribution, and of correcting for multiple testing. BB beta-binomial, FDR false discovery rate, PCA principal component analysis.

Fig. S14). Dirichlet-Multinomial-based approaches might be improved using latent space fitting, but this is not yet supported by existing implementations.

**Rare variant enrichment analysis.** We further evaluated the performance of FRASER by assessing the enrichment of rare genetic variants among splicing outlier genes based on the rationale that some aberrant splicing events are caused by rare genetic variants. For this analysis, we defined a variant as being rare when it had a minor allele frequency (MAF) less than 0.05 within GTEx[24] and gnomAD[32], as done for gene expression by Li et al.[27]. We annotated these variants in two ways. First, we considered splice region variants ("Methods" section), which were defined as variants located within 1–3 bases of an exon or 1–8 bases of an intron. This corresponds to the union of the splice site dinucleotide and splice region variants, as defined by the sequence ontology through the variant effect predictor (VEP)[33,34]. We found on average 299.4 ± 207.6 (standard deviation) rare splice region variants per sample. Second, we considered rare variants that were predicted to affect splicing by MMSplice[10], which is a machine learning algorithm that scores variants as far as 100 base pairs away from splice sites (on average 66.0 ± 48.0 rare MMSplice variants per sample; "Methods" section). The consequences of a genetic variant on splicing may spread across the splice sites of a gene because of complex effects, including competition between splice sites or coordinated splicing between distant exons[35]. Hence, the detectable effects of a variant that affects splicing are not necessarily located at its closest splice sites. Therefore, we computed the enrichment at the gene level. To this end, we computed gene-level $P$ values using a FWER correction across all splice sites within a gene ("Methods"

section). In addition to the previously benchmarked methods, we also applied the Dirichlet-Multinomial distribution-based methods LeafCutterMD[20], SPOT[21], and the LeafCutter adaptation of Kremer et al.[17].

Across all 48 GTEx tissues, FRASER showed higher enrichments than LeafCutter, Gaussian-based $P$ values, LeafCutterMD, SPOT, and non-corrected BB $P$ values. The higher enrichments observed held for different nominal $P$ value cutoffs and both for rare variants in the splice regions, as well as for those predicted to affect splicing by MMSplice (Fig. 5 and Supplementary Fig. S15). Notably, the MMSplice variant set showed 2–10 times higher enrichments across all methods compared to the splice region variant set, emphasizing the importance of considering exonic or deep intronic variants as potential splice-affecting candidates. The enrichment for MMSplice variants was even higher when computing it locally at the donor and acceptor level compared to the gene level (Supplementary Fig. S16 and Supplementary Note 1). Taken together, these benchmarks on non-simulated data confirmed the importance of controlling for covariation and using a count fraction distribution to identify aberrant splicing.

Reproducibility of outlier calls is particularly hard to assess because relevant existing datasets do not provide large amounts of replicate experiments. As a proxy for assessing the reproducibility of our calls, we investigated how often splicing outlier calls are replicated across different tissues from the same individual within GTEx. To this end, we considered outlier calls in individuals with at least 20 available tissues and on genes expressed in at least 10 tissues. This analysis revealed a surprisingly high amount of tissue-specific outlier calls for all methods (Supplementary Fig. S17), in line with the observations of Ferraro and colleagues when using SPOT on the same dataset[21]. However, FRASER had the highest percentage of outlier calls replicated across at least two

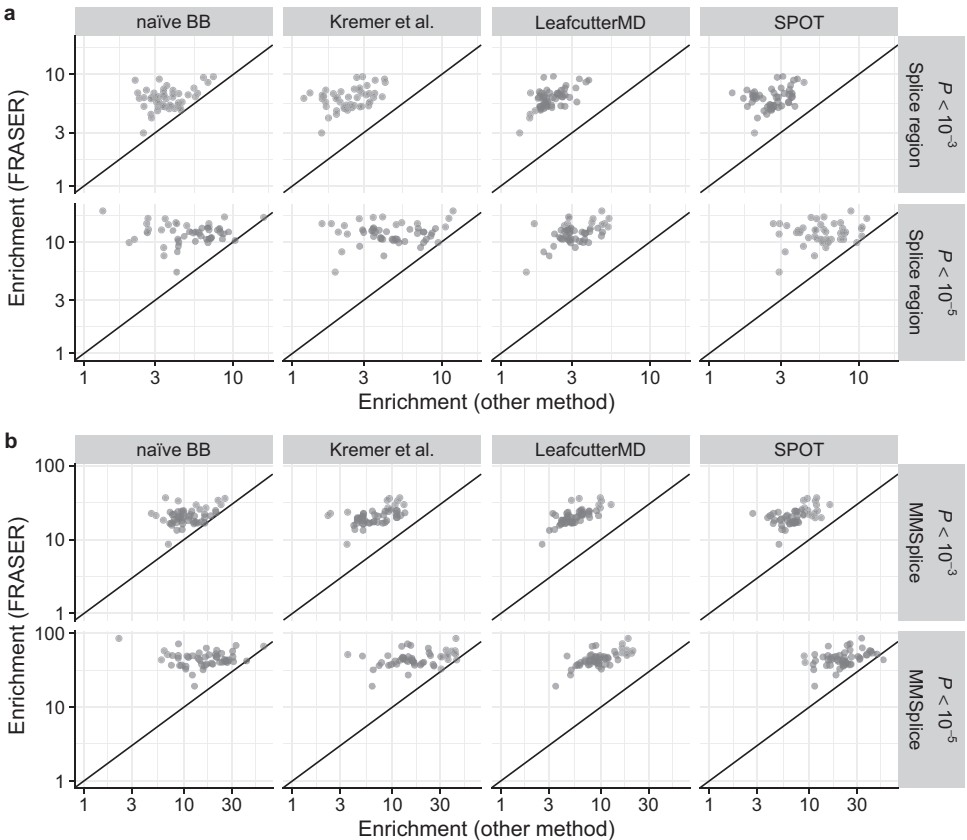

**Fig. 5 Enrichment for rare variants predicted to affect splicing. a** Enrichment using FRASER (*y*-axis) against enrichment (*x*-axis) using different aberrant splicing detection methods (columns) for rare variants located in a splice region. The enrichment is calculated for different nominal *P* value cutoffs (rows). The applied methods are a naïve beta-binomial regression, the LeafCutter adaptation of Kremer et al.[17], LeafCutterMD[20], and SPOT[21]. Each dot represents a GTEx tissue (*n* = 48). **b** Same as **a** but the enrichment is computed for rare variants predicted to affect splicing by MMSplice[10]. BB beta-binomial.

tissues among all analyzed methods (Supplementary Fig. S17). For instance, 22% of the outliers reported by FRASER at the nominal *P* value of $10^{-7}$ were also found at the nominal *P* value of $10^{-3}$ in one or more other tissues of the same individual. For, SPOT this figure is only 11%. Moreover, tissue-specific outliers were less enriched for rare variants predicted to affect splicing than outliers replicated in at least two tissues ("Methods" section, Supplementary Fig. S18). This suggests a higher proportion of false positives among the tissue-specific outliers. We have investigated the raw RNA-seq data of several such tissue-specific outliers with the Integrative Genomics Viewer (IGV)[36]. These calls actually appear to be well supported by the raw data, consistent with the fact that all methods report such an apparent excess of tissue-specific outliers. Further investigations, using biological replicates from the same tissue will be needed to understand the reason for this apparent excess of tissue-specific outliers.

**Application to rare disease diagnosis**. Having established FRASER using a large cohort of healthy donors, next we reanalyzed the 119 RNA-seq samples of skin fibroblasts from 105 individuals with a suspected rare mitochondrial disorder reported by Kremer et al. (hereafter termed the Kremer dataset)[17]. In a rare disease diagnosis context, the aim is to identify aberrant splicing events that could be disease-causing, typically by disrupting the function of a phenotypically relevant gene. Thus, gene-level statistics are handier entry points than splice site level statistics. Moreover, we suggest combining statistical significance cutoffs with effect size cutoffs, because larger effects are more likely to

have strong physiological impacts. For the Kremer dataset, FRASER reported a median of 12, 7, and 10 genes with at least one aberrant splicing event per sample for $\psi_5$, $\psi_3$, and splicing efficiency, respectively, at a significance level of FDR < 0.1 and an effect size > 0.3 (absolute difference between observed and expected value, Fig. 6a). Similar numbers of splicing outliers per sample were obtained for all 48 GTEx tissues (Supplementary Fig. S11). These criteria yielded a slightly lower number of splicing outliers than reported by the original study (1,666 versus 1,725, Fig. 6b) yet detecting all novel pathogenic splice events described in the original study (*CLPP*, *TIMMDC1* in two individuals, and *MCOLN1*, Fig. 6b). Notably, the intron retention event in the gene *MCOLN1* was missed by the aberrant splicing pipeline used by Kremer et al.[17] because it was based on LeafCutter[22], which does not consider non-split reads. (Kremer et al. identified this pathogenic event through the mono-allelic expression of a heterozygous intronic variant.) Generally, including the splicing efficiency metrics with FRASER led to a two-fold increase of detected aberrant events over considering the alternative splicing metrics $\psi_5$ and $\psi_3$ alone (Supplementary Fig. S19). Altogether, these findings show the clinical relevance and the complementarity of using both splicing efficiency and alternative splicing metrics.

Moreover, the reanalysis of the rare disease dataset highlighted aberrant alternative donor usage in the gene *TAZ* for the undiagnosed individual 74116 (difference $\psi_3 = -0.88$ and FDR $= 1.98 \times 10^{-9}$, Fig. 7a–d), which was overlooked in the original study[17]. The nearly complete loss of the canonical donor site usage of the fourth exon (Fig. 7b) resulted in the usage of a newly created donor site located 22 bp inside the fourth exon (Fig. 7e).

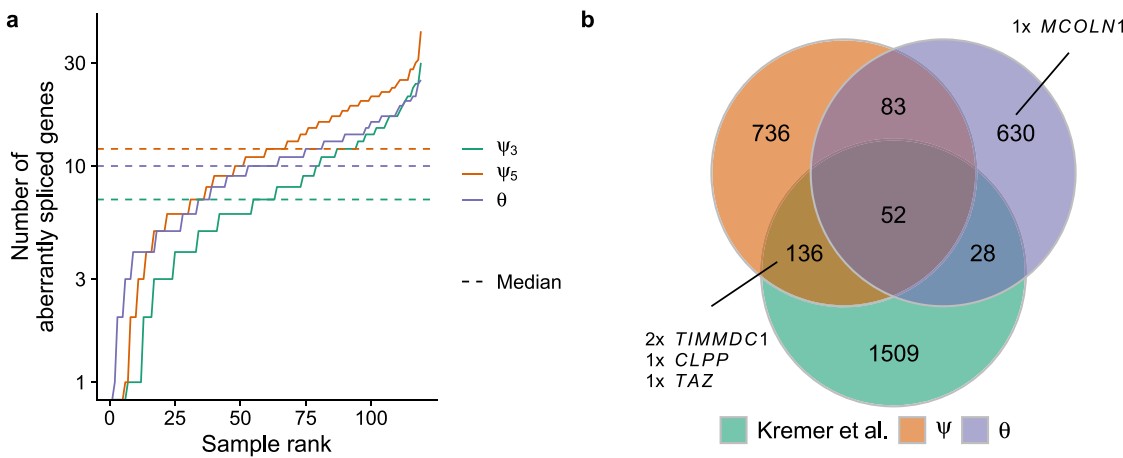

**Fig. 6 Aberrant splicing detection in a rare disease cohort. a** Number of aberrantly spliced genes within the Kremer dataset (FDR < 0.1 and |Δψ|>0.3) per sample ranked by the number of events for $\psi_5$ (orange), $\psi_3$ (green), and $\theta$ (purple). **b** Venn diagram of the aberrant splicing events detected by FRASER using alternative splicing (orange, $\psi$) or splicing efficiency (violet, $\theta$) only and detected by Kremer et al. (green)[17]. Pathogenic splicing events are labeled with the gene name. FDR false discovery rate.

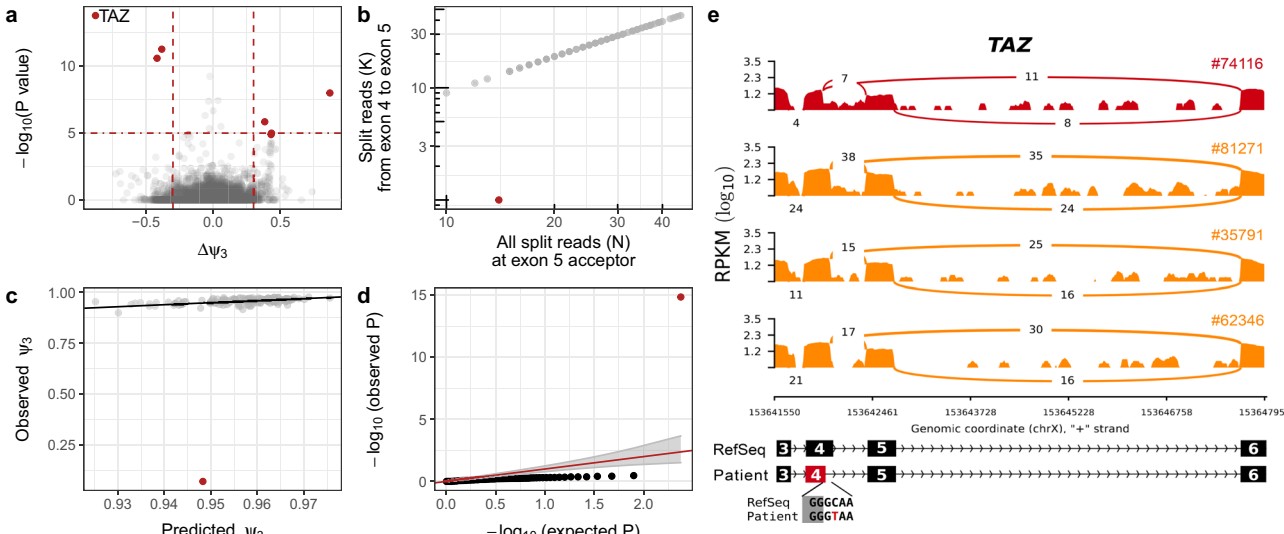

**Fig. 7 Detection of a pathogenic splicing defect using FRASER. a** Gene-level significance ($-\log_{10}(P)$, y-axis) versus effect (observed minus expected $\psi_3$, x-axis) for alternative donor usage for individual 74116. Six genes (red dots, including *TAZ*) passed both the genome-wide significance cutoff (horizontal dotted line) and the effect size cutoff (vertical dotted lines). **b** Number of split reads spanning from the fourth to fifth exon (y-axis) against the total number of split reads at the acceptor site of the fifth exon (x-axis) of the *TAZ* gene. Sample 74116 (red) deviates from the cohort trend. **c** Observed (y-axis) against FRASER-predicted (x-axis) $\psi_3$ values for the data shown in **b**. **d** Quantile-quantile plot of observed P values ($-\log_{10}(P)$, y-axis) against expected P values ($-\log_{10}(P)$, x-axis) and 95% confidence band (gray) for the data shown in **b**. Sample 74116 (red) shows an unexpectedly low P value. **e** Sashimi plot of the exon-truncation event in RNA-seq samples of the *TAZ*-affected (red) and three representative *TAZ*-unaffected (orange) individuals. The RNA-seq read coverage is given as the $\log_{10}$ RPKM-value (Reads Per Kilobase of transcript per Million mapped reads, y-axis) and the number of split reads spanning an intron is indicated on the exon-connecting line. **e** (bottom) the gene model of the RefSeq annotation is depicted in black and the aberrantly spliced exon is colored in red. The insert depicts the donor site-creating variant of the affected individual 74116. **a**, **d** P values were calculated two-sided with the beta-binomial distribution, and significance was determined based on FDR after adjusting for multiple comparisons ("Methods" section).

Usage of the new donor site leads to an ablation of eight amino acids of the protein encoded by *TAZ*, Tafazzin. Tafazzin catalyzes the maturation of cardiolipin, a major lipid constituent of the inner mitochondrial membrane that is involved in energy production and mitochondrial shape maintenance[37]. Moreover, individual 74116 harbors a rare homozygous synonymous variant (c.348C>T) that creates the new upstream donor site by introducing a GT dinucleotide (Fig. 7e). This variant had not been prioritized by WES analysis, as it was synonymous and not indexed by ClinVar[38]. However, the variant had been previously associated with a splicing defect in *TAZ* and dilated cardiomyopathy[39], consistent with the myopathic facies and arrhythmias presented by individual 74116, thereby establishing the genetic diagnosis.

The number of samples is often limited in rare disease cohorts. Hence, we investigated the sensitivity of FRASER to the sample size. To this end, we used the Kremer dataset and the 13 known pathogenic splicing events to estimate the required dataset size to reach significance for most of the clinically relevant events. We monitored the percentage of recovered pathogenic events after randomly removing samples with no pathogenic splicing defect from the dataset (Supplementary Fig. S20). As expected, the

percentage dropped with reduced sample size. With 30 samples, we recovered 85% (11 out of 13 events on average) while we needed 100 samples to capture all events regardless of the sample selection.

**Implementation**. FRASER is implemented as an R/Bioconductor package[40,41]. It contains functions to count RNA-seq reads, fit the model, calculate $P$ values, as well as to extract and visualize the results. The workflow and functionalities of the FRASER package are aligned with the previously published OUTRIDER package[29]. The package allows for a full analysis to be made with only a few lines of code and includes a comprehensive vignette that guides the user through a typical analysis step by step. It is available through Bioconductor as an open source software package (http://bioconductor.org/packages/release/bioc/html/FRASER.html). FRASER is included in the workflow Detection of RNA-seq Outlier Pipeline (DROP), which includes further analytical tools for RNA-based diagnostics[42].

## Discussion

We have introduced FRASER, an algorithm specifically developed for the detection of aberrant splicing events in RNA-seq data. The combination of three features render FRASER unique: (1) it considers non-split reads overlapping splice sites, allowing for detecting intron retention; (2) it automatically controls for latent confounders; and (3) it assesses statistical significance using a count distribution. Extensive benchmarks with artificially simulated aberrant splicing events, enrichment of rare variants with a splicing effect potential, as well as reanalysis of a rare disease cohort demonstrated the importance of each of these features. FRASER is provided as an easy-to-use R/Bioconductor package.

We implemented FRASER in a modular way so that the procedures for fitting the latent space, for estimating expected values given the latent space, as well as the distributions used to define splicing outliers  can be independently chosen. The best-performing model was obtained using a hybrid combination in which the fitting of the latent space and the estimation of the expected values are performed using a least-squared loss, while the BB distribution is used for assessing the significance of the outlier. Although this combination does not correspond to a maximum likelihood fit of a particular distribution that we are aware of, it did yield the best empirical results. Future research could investigate whether other classes of models, such as a multivariate logit-normal binomial distribution, can provide good maximum likelihood fits to the splicing metrics.

FRASER is based on splicing metrics defined at the level of individual splice sites. In theory, the use of a gene model that integrates data across entire splice isoforms can increase sensitivity because all reads supporting an isoform over another contribute to the test statistic. However, benchmarks with simulated outliers as well as enrichment analyses for rare variants predicted to affect splicing have shown that FRASER outperformed two recently described gene-level aberrant splicing methods SPOT[21] and LeafCutterMD[20]. One difficulty of gene-level methods is that either a gene model must be known beforehand or it has to be assembled de novo. In particular, SPOT appears to require robust gene models in the first place, because data preprocessing and filters yielded only 6,000 genes analyzed on a typical GTEx tissue. This filtering and clustering may be appropriate to investigate healthy populations and the basic biology of aberrant splicing[21], but limits its application in rare disease diagnostics, where any single event could be the disease-causing one. Moreover, SPOT and LeafCutterMD do not offer users the inclusion of a latent space.

Analysis of the GTEx dataset revealed a surprisingly large number of singletons, i.e., outliers called in a single sample. These singletons could be genuine tissue-specific outliers. However, the relatively lower level of enrichment for rare variants among these singletons suggests that they are enriched for false positive calls. This issue affected all investigated methods and we did not find an obvious pattern in the RNA-seq data that would raise concerns. For diagnostic applications, we advise prioritizing candidate genes by combining RNA-seq outlier calls with complementary information including genotype (presence of a rare variant) and complementary data (a replicate RNA-seq, a northern blot, or other functional assays).

FRASER has been developed to detect aberrant splicing events with a splice site-centric point of view, which is particularly adapted for short-read RNA-seq data. However, long-read sequencing technologies like Pacific Biosciences' single-molecule real-time sequencing and Oxford Nanopore Technologies' nanopore sequencing, which allow the direct sequencing of full-length transcripts[43,44], are becoming increasingly accessible. The advantage of long-read sequencing over short-read sequencing is the possibility to better assess the functional implication of defective splicing by investigating the entire sequence of the resulting isoforms. However, relative quantification of full-length isoforms using long-read sequencing remains challenging due to complex biases including 3–5′ coverage bias induced by fragmentation or pore blocking[45]. Moreover, it could also turn out that, due to the complexity of de novo transcript isoform assembly and due to the vast heterogeneity of isoforms per gene, splice site-centric approaches such as FRASER remain effective for long-read sequencing data analysis.

One limitation of the application of RNA-seq for the diagnosis of rare diseases is that the affected tissue may not be accessible. Nonetheless, a causal splicing defect may also be detectable in a clinically accessible tissue, such as blood or skin, while its pathological consequence may be revealed only in the affected tissue. The *TAZ* gene is such an example, as it has pathological effects in the heart but is nevertheless expressed in skin-derived fibroblast cells. We suggest investigators to check the gene and exon overlap with tissues of interest using the MAJIQ-CAT web interface[46]. In conclusion, based on the easy-to-use R/Bioconductor package, the integration of FRASER into DROP[42], and the advancement over alternative methods, we foresee that FRASER will become an important tool in the growing field of RNA-seq-based diagnosis of rare diseases.

## Methods

**Datasets**. We considered two RNA-seq datasets: (1) a dataset consisting of 119 RNA-seq samples from skin fibroblasts of 105 individuals with a suspected rare mitochondrial disease[17] (the Kremer dataset, https://doi.org/10.1038/ncomms15824) and (2) 7,842 RNA-seq samples from 48 tissues of 543 assumed healthy individuals of the Genotype-Tissue Expression Project V6p[24] (hereafter the GTEx dataset). The two datasets are not strand-specific. Read mapping files in the BAM file format were obtained for the Kremer dataset by mapping the RNA-seq reads to the UCSC hg19 genome assembly[47] using STAR (version 2.4.2a)[48]. To detect novel exon junctions, we ran STAR in the two-pass mode (option *twopassMode = Basic*) with a minimal chimeric segment length of 20 (*chimSegmentMin = 20*). For GTEx, we obtained the BAM files from dbGaP (phs000424.v6.p1), which were already aligned by the GTEx consortium with TopHat (version v1.4) against the GRCh37 genome assembly[49] based on the GENCODE v19 annotation[28]. We considered only samples with an RNA integrity number of 5.7 or higher and marked as usable by the GTEx consortia (SMRIN and SMAFRZE column, respectively). Further, we discarded any replicated sample and finally discarded tissues with less than 50 samples remaining.

**Read counting and splicing metrics**. The set of acceptor and donor splice sites (or the splice site map) of a dataset was defined by calling all introns, including de novo events, based on RNA-seq split reads. To this end, the split reads were extracted from the BAM files and counted using the R/Bioconductor packages GenomicAlignments and GenomicRanges[50]. Having defined the splice site map,

non-split reads overlapping splice sites were counted to compute the splicing efficiency, which can be used to detect intron retention. Specifically, the non-split reads were counted for each splice site using the R/Bioconductor Rsubread package[51] requiring at least 5 nt aligned on each side of the splice site for robustness against mapping errors of very short overhangs, as described by Braunschweig et al.[52].

As described by Pervouchine et al.[26], we compute the $\psi_5$ and $\psi_3$ values for donor $D$ (5′ splice site) and acceptor $A$ (3′ splice site), respectively, as:

$$\psi_5(D,A) = \frac{n(D,A)}{\sum_{A\prime} n(D,A\prime)} \qquad (1)$$

$$\psi_3(D,A) = \frac{n(D,A)}{\sum_{D\prime} n(D\prime,A)}, \qquad (2)$$

where $n(D, A)$ denotes the number of split reads spanning the intron between donor $D$ and acceptor $A$ and the summands in the denominators are computed over all acceptors that spliced with the donor of interest (Eq. (1)) and all donors that spliced with the acceptor of interest (Eq. (2)). To not only detect alternative splicing but also partial or full intron retention, we considered a splicing efficiency metric. Multiple related definitions exist including 3′ splice site ratio[53], completeness of splicing index[54], and percent intron retained[52]. We used the $\theta_5$ and $\theta_3$ values as defined by Pervouchine et al.[26]. Specifically:

$$\theta_5 = \frac{\sum_{A\prime} n(D,A\prime)}{n(D) + \sum_{A\prime} n(D,A\prime)} \qquad (3)$$

$$\theta_3 = \frac{\sum_{D\prime} n(D\prime,A)}{n(A) + \sum_{D\prime} n(D\prime,A)}, \qquad (4)$$

where $n(D)$ is the number of non-split reads spanning the exon-intron boundary of donor $D$, and $n(A)$ is defined as the number of non-split reads spanning the intron-exon boundary of acceptor $A$. While calculating $\theta$ for the 5′ and 3′ splice sites separately, $\theta_5$ and $\theta_3$ were not distinguished later in the modeling step and hence, we termed it jointly $\theta$ in the remainder of the manuscript.

For robust fitting of the model, we restricted the analysis to splice sites of introns supported by at least 20 split reads in at least one sample. Further, we kept only splice sites and introns with at least one read coverage in 95% of the samples.

**Statistical model.** The metrics $\psi_5$, $\psi_3$, and $\theta$ are count proportions. For each of these metrics, we model the distribution of the numerator conditioned on the value of the denominator using the BB distribution. Unlike the binomial distribution, the BB distribution can account for overdispersion. Specifically, for $\psi_5$, we assume that the split read count $k_{ij}$ of the intron $j = 1, …, p$ in sample $i = 1, …, N$ follows a BB distribution with an intron-specific intra-class correlation parameter $\rho_j$ and a sample- and intron-specific proportion expectation $\mu_{ij}$:

$$P(k_{ij}) = BB(k_{ij}|n_{ij}, \mu_{ij}, \rho_j), \qquad (5)$$

where $n_{ij}$ defines the total number of split reads having the same donor site than intron $j$. The metrics $\psi_3$ and $\theta$ are modeled analogously. For ease of writing, we will refer in the following with $\psi$ always to the site-specific $\psi_5$ and $\psi_3$ form. Both $\mu_{ij}$ and $\rho_j$ are limited to the range [0,1]. The parametrization of the BB distribution used here can be found in the Supplementary Note 3.

The proportion expectation $\mu_{ij}$ is jointly modeled using a latent space that captures covariations between samples. Specifically, we model:

$$\mu_{ij} = \sigma(y_{ij}) = \frac{\exp(y_{ij})}{1 + \exp(y_{ij})}, \qquad (6)$$

$$\mathbf{y}_i = \mathbf{h}_i \mathbf{W}_d + \mathbf{b}, \qquad (7)$$

$$\mathbf{h}_i = \bar{\mathbf{x}}_i \mathbf{W}_e, \qquad (8)$$

where the vectors $\mathbf{h}_i$ are the rows of the matrix $\mathbf{H}$, the $N \times q$ projection of the data onto the $q$-dimensional latent space with $1 < q < \min(p, N)$, $\mathbf{W}_e$ is the $p \times q$ encoding matrix, $\mathbf{W}_d$ is the $q \times p$ decoding matrix, and the $p$-vector $\mathbf{b}$ is a bias term. The input row vector $\bar{\mathbf{x}}_i$ is given by the centered logit-transformed pseudocount ratios. We define $\bar{\mathbf{x}}_i$ as:

$$\bar{x}_{ij} = x_{ij} - \bar{x}_j, \qquad (9)$$

$$x_{ij} = logit\left(\frac{k_{ij} + 1}{n_{ij} + 2}\right), \qquad (10)$$

$$logit(a) = \log \frac{a}{1-a}. \qquad (11)$$

**Fitting of the latent space and the distribution.** Four parameters must be fitted, namely $\mathbf{W}_e$ (the encoding matrix), $\mathbf{W}_d$ (the decoding matrix), $\mathbf{b}$ (the bias term), and $\rho_j$ (the intra-class correlation of the BB distribution). The fitting of these parameters is achieved in two steps. First, the latent space $\mathbf{H}$ and the expected splicing

proportions $\mu_{ij}$ are fitted using a PCA. To this end, a PCA is computed on the input matrix $\tilde{\mathbf{X}}$ using the pcaMethods package[55]. The latent space $\mathbf{H}$ is then computed using Eq. (8) by setting the encoder matrix $\mathbf{W}_e$ to the first $q$ loadings of the PCA. Given the latent space $\mathbf{H}$, $\mu_{ij}$ is computed using the transpose of $\mathbf{W}_e$ for $\mathbf{W}_d$ and setting the bias term to $\bar{\mathbf{x}}_j$. In the second step, the intra-class correlation parameters of each intron $j$, $\rho_j$, are fitted given the count proportion expectations using a BB loss function. Specifically, we use the *optimize* function from R[41] and minimize the average negative BB log-likelihood in parallel across introns (Supplementary Note 3).

**Alternative distribution fitting using a beta-binomial regression.** Moreover, we implemented an alternative approach to fit the distribution parameters given the latent space $\mathbf{H}$. To this end, we use a negative BB log-likelihood loss function to model in an iterative fashion the decoding matrix $\mathbf{W}_d$ and the bias term $\mathbf{b}$ on the one hand and the intra-class correlation parameter $\rho$ on the other hand. First, we initialize the parameters as described above using PCA. Given the latent space, these parameters can be fitted independently for each intron $j$. We start by optimizing $\rho_j$ given the decoder coefficients $\mathbf{w}_j^d$ and the bias $b_j$ (step 1). Subsequently, we optimize $\mathbf{w}_j^d$ and $b_j$ given $\rho_j$ in step 2. Steps 1 and 2 are repeated until the average negative log-likelihood of each step in one iteration does not differ by more than the convergence threshold of $10^{-5}$ from the last step of the previous iteration, or until 15 iterations are reached, which triggers a warning. We use the L-BFGS method implemented in the R function *optim* to fit the decoder coefficients and the bias[56]. A detailed derivation of the loss functions and the respective gradients can be found in the Supplementary Note 3.

**Alternative distribution fitting using a robust beta-binomial regression.** Outlier data points can have strong effects on the BB regression. We therefore also implemented an alternative method based on a robust BB regression. To this end, we used weights in the loss function to decrease the influence of outliers on the BB regression, according to the edgeR approach[57]. Specifically, we defined the weight for each observation based on its Pearson residual. The Pearson residual ($r_{ij}$) of the observed data point $x_{ij}$ (Eq. (10)) with respect to the BB distribution including the pseudocounts is defined as follows:

$$r_{ij} = \frac{observed - expected}{\sqrt{Var(expected)}} = \frac{x_{ij} - \mu_{ij}}{\sqrt{\frac{\mu_{ij}(1-\mu_{ij})(1+(n_{ij}-1)\rho_j)}{n_{ij}+2}}}. \qquad (12)$$

The weights $w_{ij}$ for sample $i$ and intron $j$ are obtained from these residuals using the Huber function[58]:

$$w_{ij} = \begin{cases} 1 & \text{for } |r_{ij}| \le k, \\ \frac{k}{|r_{ij}|}, & \text{otherwise} \end{cases}, \qquad (13)$$

where we use $k = 1.345$ as suggested in the edgeR package[57], which leads to the downweighting of about 5% of the data points. These weights are then included in the calculation of the negative log-likelihood yielding the average weighted negative log-likelihood $L^W$:

$$L^W = \frac{1}{p \times N} \sum_{i,j} w_{ij} L_{ij}, \qquad (14)$$

$$L_{ij} = -\log\left(BB(k_{ij}|n_{ij}, \mu_{ij}, \rho_j)\right), \qquad (15)$$

where $L_{ij}$ is the negative BB log-likelihood of sample $i$ and intron $j$ as defined in the Supplementary Note 3.

**Finding the hyperparameter.** All three model fitting procedures described above leave one hyperparameter that requires optimization: the latent space dimension $q$. To find the optimal latent space dimension $q$, we implemented a denoising auto-encoder approach[23]. Specifically, we generated corrupted data by injecting aberrant read count ratios with a frequency of $10^{-2}$ into the original data. The injection scheme is laid out in detail in the next section. We then select $q$ as the dimension maximizing the area under the precision-recall curve for identifying the corrupted read ratios. This is done for each splicing metric separately. To speed up the fitting procedure of the hyperparameter, we randomly subset the input matrix $\tilde{\mathbf{X}}$ to 15,000 introns out of the 30,000 most variable introns with a mean total coverage greater than 5. This subsetting is performed before the injection of aberrant read count ratios.

**In silico injection of artificial outliers.** To fit the FRASER hyperparameter as well as to compare the outlier detection performance between FRASER and other methods, we developed a procedure to inject artificial outliers into a given dataset. For injection, we considered all expressed introns or splice sites within the dataset and injected only one outlier per splice site and sample. Outliers were randomly injected with a frequency of $10^{-2}$ for the hyperparameter optimization and with a frequency of $10^{-3}$ for the benchmarking.

To create aberrant splicing ratios, we inject in silico a splicing outlier count $k_{ij}^o$ by changing the original read count $k_{ij}$ such that the value of $\psi_{ij}$ changes by $\Delta\psi_{ij}^o$. $\Delta\psi_{ij}^o$ is derived from a uniform distribution:

$$\Delta\psi_{ij}^o \sim \pm\, \mathrm{U}(0.2, \Delta\psi_{ij}^{max}), \tag{16}$$

where $\Delta\psi_{ij}^{max}$ is the maximal possible $\Delta\psi_{ij}$ for intron $j$ in sample $i$. The value of $\Delta\psi_{ij}^{max}$ is dependent on the randomly sampled injection direction: $\Delta\psi_{ij}^{max} = 1 - \psi_{ij}$ and $\Delta\psi_{ij}^{max} = \psi_{ij}$ for up- or down-regulation, respectively. To ensure that an aberrant splice ratio can be injected the direction is switched if $\Delta\psi_{ij}^{max} < 0.2$. We injected outliers only for introns harboring 10 reads or more in the considered sample.

Taking the pseudocounts into account, the outlier count $k_{ij}^o$ is then given by

$$k_{ij}^o = \mathrm{round}((\psi_{ij} \pm \Delta\psi_{ij}^o) \cdot (n_{ij} + 2) - 1). \tag{17}$$

In order to provide a biologically realistic outlier injection scheme that preserves the total amount of reads, the counts for the introns $l$ sharing the same donor or acceptor, respectively, with $k_{ij}^o$ are changed accordingly, where the $\Delta\psi_{ij}^o$ change is distributed equally over all secondary introns $l$, as follows:

$$\Delta\psi_{il}^s = -\Delta\psi_{ij}^o \cdot \frac{\psi_{il}}{1 - \psi_{ij}} \tag{18}$$

$$k_{il}^s = \mathrm{round}((\psi_{il} \pm \Delta\psi_{il}^s) \cdot (n_{il} + 2) - 1). \tag{19}$$

**Injection of splicing outliers by interchanging reads between tissues.** Benchmarking FRASER against Dirichlet-Multinomial distribution-based algorithms required an alternative injection scheme. To this end, we swapped reads mapping to alternatively spliced genes between two GTEx tissue samples of the same individual. We used LeafCutter[22] with the default parameters to detect alternatively spliced genes between the suprapubic skin tissue and the brain cortex tissue and used 40 random individuals sequenced in both tissues. We then randomly selected 60 gene-sample pairs from the top 100 LeafCutter hits and replaced all reads falling into the gene body of the given suprapubic skin tissue sample with the reads from the brain cortex tissue sample. The benchmark was then performed on the 40 suprapubic skin tissue samples previously selected.

**Statistical significance.** The statistical significance of outliers is assessed by testing the null hypothesis that the count $k_{ij}$ with $n_{ij}$ trials follows a BB distribution with parameters fitted as described above for every pair of sample $i$ and intron $j$. We compute two-sided $P$ values $p_{ij}$ using the mean probability of success $\mu_{ij}$ and the fitted intra-class correlation parameter $\rho_j$, as follows:

$$p_{ij} = 2 \cdot \min\left\{\frac{1}{2}, \sum_{k=0}^{k_{ij}} \mathrm{BB}(k|n_{ij}, \mu_{ij}, \rho_j), 1 - \sum_{k=0}^{k_{ij}-1} \mathrm{BB}(k|n_{ij}, \mu_{ij}, \rho_j)\right\}. \tag{20}$$

The term ½ is included to prevent the generation of $P$ values greater than 1, which can happen due to the nature of the discrete distributions.

The $P$ values of introns sharing a splice site are not independent, as the sum of the proportions on which they are based is one. Therefore, we correct the $P$ values for each splice site with the FWER using Holm's method, which holds under arbitrary dependence assumptions[30], and report the minimal corrected $P$ value per splice site. An additional FWER step is performed at the gene level if gene-level $P$ values are requested. To correct for multiple testing genome-wide, we use the Benjamini–Yekutieli FDR method[31] as both splice site-corrected $P$ values and the gene-wise corrected $P$ values can still be correlated due to biological effects that are not completely removed by the model. All $P$ value corrections are performed on a per-sample basis.

**Z score and $\Delta\psi$ calculation.** Z scores $z_{ij}$ are calculated per intron on the difference on the logit scale between the measured $\psi$ value including pseudocounts and the proportion expectation $\mu_{ij}$, as follows:

$$z_{ij} = \frac{\delta_{ij} - \bar{\delta}_j}{\mathrm{sd}(\delta_j)}, \tag{21}$$

$$\delta_{ij} = \mathrm{logit}\left(\frac{k_{ij} + 1}{n_{ij} + 2}\right) - \mathrm{logit}(\mu_{ij}). \tag{22}$$

$$\mathrm{logit}(a) = \log\left(\frac{a}{1 - a}\right). \tag{23}$$

The $\Delta\psi$ values are calculated as the difference between the observed $\psi_{ij}$ value on the natural scale including pseudocounts and the proportion expectations $\mu_{ij}$:

$$\Delta\psi_{ij} = \psi_{ij} - \mu_{ij} = \frac{k_{ij} + 1}{n_{ij} + 2} - \mu_{ij}. \tag{24}$$

**Alternative splicing outlier detection methods.** We implemented different alternative splicing outlier detection methods to assess the performance of FRASER. As the baseline for our approach, we used a simple BB distribution with no correction for existing covariation and the parameters $\mu_{ij}$ and $\rho_j$ were estimated using the VGAM package in R[59]. Further, we implemented a $z$ score approach similar to the approach described by Frésard et al.[18]. Instead of regressing out the top $q$ principal components accounting for 95% of the variation within the data, we used the top $q$ loadings of the PCA maximizing the precision-recall of in silico injected splicing outliers and computed the $z$ scores according to Eq. (21). Furthermore, we implemented three Dirichlet-Multinomial distribution-based methods. First, the LeafCutter[22] approach described by Kremer et al.[17], in which one sample is compared against all others within the dataset and no control for latent sources of sample covariation is considered. Second, the LeafCutterMD method[20], which is an advancement of LeafCutter by modeling splicing outliers directly. And third, the SPOT approach[21], which uses Mahalanobis distance-based empirical $P$ values to capture splicing outliers. All three approaches were run with their default parameters.

**Enrichment analysis.** For the GTEx enrichment analysis, we obtained all rare variants (MAF < 0.05 within all 635 GTEx samples and in gnomAD[32]) from the GTEx whole-genome sequencing genetic variant data (V6p)[24]. From this rare variant set, we extracted all annotated splicing variants (*splice_donor*, *splice_acceptor*, and *splice_region*) according to the sequencing ontology and VEP[33,34]. This covers all variants surrounding the exon-intron and intron-exon boundary, which is 1–3 bases within the exon and 1–8 bases within the intron. We also extracted variants predicted to affect splicing by MMSplice[10]. To this end, we scored all variants within 100 bp of an annotated exon (GENCODE release 30[28]) using MMSplice in an exon-centric way. Subsequently, variant-exon pairs with a score of $|\Delta\mathrm{logit}(\psi)| > 2$ were selected. We then computed enrichments for rare splicing variants found within outlier genes as the proportion of outliers having a rare splicing variant over the proportion of non-outliers having a rare splicing variant.

**Enrichment analysis of reproducible splicing outliers.** The enrichment analysis of reproducible splicing outliers was performed on the gene level and on the variant sets described above. Before computing the enrichment, we filtered down our splicing outlier call set. First, we selected any individual from the GTEx dataset that was sequenced in at least 20 tissues (195 individuals remained). From this subset, we only considered individual-gene pairs that were tested and hence passed the expression filters as described in the counting section in at least 10 tissues. Based on this subset, we call a splicing outlier in a given tissue reproducible if it is detected in one or more other tissues at a nominal $P$ value $<10^{-3}$. The enrichment was then computed on the full dataset but only with reproducible splicing outlier calls.

**Reporting summary.** Further information on research design is available in the Nature Research Reporting Summary linked to this article.

## Data availability
The GTEx dataset is available through dbGaP (accession number: phs000424.v6.p1). The same sequencing data as described by Kremer et al.[17] was used (https://doi.org/10.1038/ncomms15824). Split-read data for the Kremer et al. dataset[17] produced in this study can be accessed through Zenodo: https://doi.org/10.5281/zenodo.4271599.

## Code availability
The FRASER package developed in this study is released on Bioconductor: https://doi.org/10.18129/B9.bioc.FRASER. The analysis pipeline used throughout this study can be accessed through GitHub at: https://github.com/gagneurlab/FRASER-analysis.

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

## Acknowledgements

We thank Joseph Aicher for comments on the manuscript. C.M., V.A.Y., H.P., and J.G. were supported by the EU Horizon 2020 Collaborative Research Project SOUND (633974). The Bavaria California Technology Center supported C.M. through a fellowship. The German Bundesministerium für Bildung und Forschung (BMBF) supported the study through the e:Med Networking fonds AbCD-Net (FKZ 01ZX1706A to V.A.Y., C.M., and J.G.), the German Network for Mitochondrial Disorders (mitoNET; 01GM1113C to H.P.), and the E-Rare project GENOMIT (01GM1207 to H.P.). M.H.Ç. was supported by the Competence Network for Technical, Scientific High Performance Computing in Bavaria KONWIHR. The Genotype-Tissue Expression (GTEx) Project was supported by the Common Fund of the Office of the Director of the National Institutes of Health and by the National Cancer Institute, National Human Genome Research Institute, National Heart, Lung, and Blood Institute, National Institute on Drug Abuse, National Institute of Mental Health, and National Institute of Neurological Disorders and Stroke. The data used for the analyses described in this manuscript were obtained from the GTEx Portal on April 4, 2019, under accession number dbGaP: phs000424.v6.p1.

## Author contributions

C.M. and J.G. conceived the method with the help of I.S. C.M. and I.S. implemented the package and performed the full analysis. V.A.Y. contributed to the package development and to the analysis. M.H.Ç. performed the MMSplice analysis of GTEx. C.M. and Y.L.

performed the rare variant enrichment analysis. L.S.K. and M.G. analyzed the results of the rare disease cohort. J.G and H.P. supervised the research. C.M., I.S, and J.G. wrote the manuscript with the help of V.A.Y. All authors revised the manuscript.

## Funding

## Competing interests

The authors declare no competing interests.
