## [Peer Review File · Nature Communications]

Reviewer #1 (Remarks to the Author):

The authors develop a method, FRASER, to detect splicing outliers from short read RNA-seq. Alternative donor and acceptor choice is modeled, as well as intron retention by considering non-split reads overlapping splice sites. A per tissue linear denoising autoencoder is learnt on logit-transformed ratios, and statistical deviation from that model is measured under a separately fit beta-binomial (with per junction overdispersion parameters). Promising results are presented on GTEx using artificially injected outliers and on the Kremer et al dataset.

The paper is well written apart from a few typos: many of these are caught by the word spell checker! Personally I would have little more detail on the methods in the intro to results (e.g. that it's a linear autoencoder fit using PCA, and that you use a per junction dispersion parameter for the BB, but still no equations).

My main concern with the paper is the splice-site centric view. One of the main forms of alternative splicing is exon skipping. While this is represented indirectly by splice site choice it is clearly statistically inefficient to test the splice sites involved separately. A Dirichlet-multinomial approach would be one way to address this. Related to this is the issue that the main quantitative comparison is done simulating data that fits the FRASER assumptions very well: only one junction's usage is modified independently of others (apart from normalization). Maybe a more "realistic" way to generate splicing outliers would be to set the counts for one gene/individual to those for a different tissue where the gene is differentially spliced (relative to the tissue currently under consideration). Code for the DM approach is now available here: <https://github.com/BennyStrobes/SPOT>
So it would be very helpful to see a comparison to this for such an experiment.

For the clustering of GTEx tissues I need to be convinced this isn't driven by total expression changes. The normalization using pseudocounts used means that for very low ratio junctions as total expression increases the measured ratio will get smaller as the pseudocounts become less important. With the logit transformation that could potentially have a large influence.

Minor comments

- "This typically doubles the number of detected aberrant events and identified a pathogenic intron retention in MCOLN1" in the abstract: in what disease?
- "collectively"
- 2D: presumably you could also be removing biological variation which might be bad? Or maybe that's also helpful here?
- "an effect size < 0.3" you mean >
- equation 14: why minus?
- why calculate z scores on the logit scale?

Reviewer #2 (Remarks to the Author):

Mertes et al. describe a new analytical framework for detecting unusual splicing in RNA-seq. There is increasing demand for looking at new clues for unsolved disease and splicing variation ranks high among changes potentially missed by interpretation of disease exomes or genomes based on sequence alone. Motivation here is to apply more rigorous statistical framework to allow confidence for outlier detection and to reduce noise that plagued earlier methods. As far as statistics go, their approach is clearly increasing specificity – however, in case of rare disease discovery (particularly for cases that are missed by traditional analyses) there is natural tendency to maximize sensitivity to build hypotheses – therefore I think the assessment of these methods is always context specific and requires in most cases joint analyses of phenotype, genotype and functional genomics. Less is not always more in unsolved rare disease. Few specific comments below:

- The 60% figure of splicing prevalence does (intro first sentence) begs for clarifying in which context such extreme figure would be reasonable, certainly not across all Mendelian disease.

- The simulation framework to measure recall falls short of needed validation (as does demonstration of few previously detected rare variants). Similarly, showing enrichment for rare variation gives little idea of $n=1$ situation, where one needs to call an outlier in a non-diagnostic sample. My suggestion would be to utilize the multitissue nature of GTEx – as authors appropriately identify many non-canonical splice events and suggest potential rare aberrant splicing events driven by genetic variation. Clearly then if the method works these rare events would be shared in independent tissue samples from same individual. I acknowledge that some events are tissue specific, but seeing no enrichment of same rare splice event identification would raise alarms. Furthermore, such tissue replicated rare splice events should show whopping enrichments for rare variants in splice regions. I think the paper is already narrow in scope and performing full analyses on public datasets is warranted as no new data is presented.

- In discussion it would be useful to reflect to whole transcript sequencing methods that are becoming increasingly accessible with lowering costs of long-read sequencing (e.g. PacBio IsoSeq).

Minor comments:

- There is one hyperparameter to be fit (the latent space dimension), but it seems like there are a number of choices that could affect performance (weighting method in Eq 12, distribution of artificial outlier counts in Eq 15, etc).

- Eq 19. The sum is integrating under the discrete probability distribution, right? If so, I missed why the $\frac{1}{2}$ is needed? Shouldn't the $\text{sum}_k(\text{BB})$ part be in the range $[0,1]$, so that either $\text{sum}_k(\text{BB}) < 1/2$ or $1 - \text{sum}_k(\text{BB}) < 1/2$?

Point to point response (NCOMMS-20-04479)

Reviewer #1 (Remarks to the Author):

The authors develop a method, FRASER, to detect splicing outliers from short read RNA-seq. Alternative donor and acceptor choice is modeled, as well as intron retention by considering non-split reads overlapping splice sites. A per tissue linear denoising autoencoder is learnt on logit-transformed ratios, and statistical deviation from that model is measured under a separately fit beta-binomial (with per junction overdispersion parameters). Promising results are presented on GTEx using artificially injected outliers and on the Kremer et al dataset.

The paper is well written apart from a few typos: many of these are caught by the word spell checker! Personally I would have little more detail on the methods in the intro to results (e.g. that it's a linear autoencoder fit using PCA, and that you use a per junction dispersion parameter for the BB, but still no equations).

Response:

We have added a sentence in the third paragraph of the introduction stating that the latent space is obtained by PCA. We added a further sentence in the Results subsection "Calling aberrant splicing events using the beta-binomial distribution" explaining that the junctions are modelled independently of each other. In addition, we ran Google's Grammarly to correct typos.

My main concern with the paper is the splice-site centric view. One of the main forms of alternative splicing is exon skipping. While this is represented indirectly by splice site choice it is clearly statistically inefficient to test the splice sites involved separately. A Dirichlet-multinomial approach would be one way to address this.

Response:

Exon skipping is indeed one of the main forms of alternative splicing regulation. However, to our knowledge, it is unclear whether exon skipping is one of the main forms of aberrant splicing, or one of the main forms of pathogenic aberrant splicing. Nonetheless, statistical power could in principle be gained for detecting aberrant exon skipping by considering a multivariate approach across multiple splice sites. However, moving from a univariate to a multivariate distribution comes at a price, in this case building a splicegraph before testing. This adds complexity to the overall approach and therefore can lead to a loss of robustness. We are now comparing FRASER with two novel aberrant splicing detection methods that use a Dirichlet-multinomial approach: SPOT, a not yet peer-reviewed method suggested by this reviewer in the subsequent point, and LeafCutterMD, another

method that has been published while our manuscript was under review. These new benchmarks show that FRASER outperforms these approaches (See below).

Related to this is the issue that the main quantitative comparison is done simulating data that fits the FRASER assumptions very well: only one junction's usage is modified independently of others (apart from normalization). Maybe a more "realistic" way to generate splicing outliers would be to set the counts for one gene/individual to those for a different tissue where the gene is differentially spliced (relative to the tissue currently under consideration). Code for the DM approach is now available here: <https://github.com/BennyStrobes/SPOT>. So it would be very helpful to see a comparison to this for such an experiment.

Response:

Based on this suggestion, we have implemented a benchmark method that allows a fairer comparison with existing DM-based approaches: LeafCutter as adapted by Kremer et al. and two new methods, SPOT, as suggested, and the recently published LeafCutterMD. Specifically, we added a new benchmark that introduces splicing outliers by swapping out, for a single individual, skin and brain tissue read counts for an entirely differentially alternatively spliced gene. These differentially spliced genes were identified by LeafCutter, a DM-based method, giving a potential advantage to DM-based aberrant splicing detection methods. Nevertheless, precision-recall curves for this simulation setting shows that FRASER outperforms the DM-based methods (new Supplementary Figure S14). Moreover, these new simulation-based results are consistent with the stronger enrichment among FRASER reported events for rare variants predicted to affect splicing, an evidence that is not based on any simulation assumption (Supplementary Figure S15 updated with LeafCutterMD and SPOT). DM-based approaches might be improved using latent space fitting, but this is not yet supported by SPOT and LeafCutterMD. The discussion now touches on this point.

For the clustering of GTEx tissues I need to be convinced this isn't driven by total expression changes. The normalization using pseudocounts used means that for very low ratio junctions as total expression increases the measured ratio will get smaller as the pseudocounts become less important. With the logit transformation that could potentially have a large influence.

Response:

The sample correlation analysis in Figure 2 is not as sensitive to total coverage as pointed out by the reviewer because, for this analysis, we are clipping splicing metrics to the [0.01, 0.99] interval. We now explain this in the legend of Figure 2. To investigate a potential influence of pseudocounts on sample clustering, we have now computed the sample correlations in a pairwise fashion on introns that are expressed in both samples for different minimal coverage cutoffs (e.g.: $N \geq 100$), where pseudocount effects should be minimized. Even when only considering introns with a pairwise coverage of more than 100, which are only a small subset out of the total observed introns, we still observe

sample correlations. These are actually even stronger (Supplementary Figure S4 and new sentence in the second paragraph in the Results section). From this analysis, we conclude that the clustering that we observed in Figure 2 is not an artefact driven by expression levels and caused by pseudo-counts, but represents covariations in the splicing metrics that are present in the GTEx data. This co-variation can have a biological ground such as variation in trans of splicing factors and pathways.

Minor comments

- **"This typically doubles the number of detected aberrant events and identified a pathogenic intron retention in MCOLN1" in the abstract: in what disease?**

Response:

We adapted it now to: identified a pathogenic intron retention in MCOLN1 causing mucopolipidosis.

- **"Collectievly"**

Response:

It is corrected now.

- **2D: presumably you could also be removing biological variation which might be bad? Or maybe that's also helpful here?**

Response:

Indeed latent space fitting removes technical and biological variation (e.g. due to sex or common genetic variations) that frequently affect splicing statistics across multiple individuals. The goal of our method, which is developed for rare disease diagnostics, is to detect rare, individual-specific deviations from these common variations. To this end, latent space fitting is helpful.

- **"an effect size < 0.3" you mean >**

Response:

It is now corrected.

- **equation 14: why minus?**

Response:

We actually use the negative log likelihood instead of log-likelihood. Hence, we had a minus in front of Equation 14. We now corrected the text of this subsection of the Methods.

- **why calculate z scores on the logit scale?**

Response:

As our model predicts the expected values on the logit scale, we chose to also compute the z scores on the logit scale. The z scores from the two scales are strongly correlated. However, as there are some differences, we have now added an additional comparison with z scores on the natural scale to be comparable to the method used by Frésard et al. (Supplementary Information and new Supplementary Figure S8). The results show that changing the scale on which z scores are calculated has some effect on the precision recall curves, but FRASER still outperforms both z score based approaches.

Reviewer #2 (Remarks to the Author):

Mertes et al. describe a new analytical framework for detecting unusual splicing in RNA-seq. There is increasing demand for looking at new clues for unsolved disease and splicing variation ranks high among changes potentially missed by interpretation of disease exomes or genomes based on sequence alone. Motivation here is to apply more rigorous statistical framework to allow confidence for outlier detection and to reduce noise that plagued earlier methods. As far as statistics go, their approach is clearly increasing specificity – however, in case of rare disease discovery (particularly for cases that are missed by traditional analyses) there is natural tendency to maximize sensitivity to build hypotheses – therefore I think the assessment of these methods is always context specific and requires in most cases joint analyses of phenotype, genotype and functional genomics. Less is not always more in unsolved rare disease. Few specific comments below:

- The 60% figure of splicing prevalence does (intro first sentence) begs for clarifying in which context such an extreme figure would be reasonable, certainly not across all Mendelian disease.**

Response:

The 60% figure stemmed from an early estimation. We are now providing the 15%-30% intervals and are referring to more recent publications.

- The simulation framework to measure recall falls short of needed validation (as does demonstration of few previously detected rare variants). Similarly, showing enrichment for rare variation gives little idea of n=1 situation, where one needs to call an outlier in a non-diagnostic sample. My suggestion would be to utilize the multi tissue nature of GTEx – as authors appropriately identify many non-canonical splice events and suggest potential rare aberrant splicing events driven by genetic variation. Clearly then if the method works these rare events would be shared in independent tissue samples from the same individual. I acknowledge that some events are tissue specific, but seeing no**

enrichment of the same rare splice event identification would raise alarms. Furthermore, such tissue replicated rare splice events should show whopping enrichments for rare variants in splice regions. I think the paper is already narrow in scope and performing full analyses on public datasets is warranted as no new data is presented.

Response:

We have now added an analysis of the reproducibility of splicing outlier calls across GTEx tissues. GTEx has only a handful of replicated RNA-seq samples. Hence, it is hard to conclude whether a sample-specific event is a genuine tissue-specific event or a non-replicable observation. We agree with the reviewer that singletons, i.e. calls that are found in a single tissue and are not replicated in any other, are probably dubious. To perform this analysis, we considered outlier calls in individuals with at least 20 available tissues and on genes expressed in at least 10 tissues. All methods report an excess of singletons (new Supplementary Figure S16). This observation is in line with the findings in Ferraro et al (bioRxiv 2019). We note that FRASER has the lowest percentage of singletons among all analysed methods (Supplementary Figure S16). For instance, 22% of the outliers reported by FRASER at the nominal P value of 10^{-7} are found at the nominal P value of 10^{-3} in one or more other tissues of the same individual. For SPOT, this figure is only 11%. Moreover, singletons are less enriched for rare variants predicted to affect splicing than outliers replicated in at least two tissues (Methods, Supplementary Figure S17). This suggests a higher amount of false positives among the singletons. We have investigated the raw data for several such singletons with the visualization tool IGV. These singletons actually appear to be well supported by the raw data. We could not find an obvious recurrent pattern that would explain them. This is consistent with the fact that all methods report such an apparent excess of singletons. Further investigations, using proper replicate experiments are needed to understand the reason for this apparent excess of singletons and potentially to provide criteria to filter them out. We now warn the reader of this issue in discussion. For diagnostic applications, we advise to base a prioritization with complementary information including genotype (presence of a rare variant) and complementary data (a replicate RNA-seq, a northern blot, or functional assays).

- **In discussion it would be useful to reflect on whole transcript sequencing methods that are becoming increasingly accessible with lowering costs of long-read sequencing (e.g. PacBio IsoSeq).**

Response:

We added a new paragraph in the discussion addressing whole transcript sequencing methods.

Minor comments:

- **There is one hyperparameter to be fit (the latent space dimension), but it seems like there are a number of choices that could affect performance (weighting method in Eq 12, distribution of artificial outlier counts in Eq 15, etc).**

Response:

We had investigated the effect of the weighting scheme. The weighting scheme is based on a classical loss function from the robust statistics literature, the Huber estimator. The parameter k of the Huber estimator is typically set to 1.345 in normally distributed settings to achieve 95% efficiency [Fox J. Robust Regression. Behav. Res. Methods. 2002]. We relied on this often-used value of k (also used by edgeR) and did not investigate alternative values. The weighting scheme turns out to not be useful when fitting the latent space. It was helpful to regress the individual beta-binomial distributions against the latent space but at a prohibitive computing cost compared to least square regression with a comparable performance. Hence, it did not end up in the final implementation of FRASER. We edited the methods to make this clearer.

The only other parameters FRASER relies upon concern the distribution of artificial outlier counts. We had investigated various modes of injecting artificial outliers but did not report these investigations in the original submission, leaving the impression of a somewhat arbitrary modeling choice. The results of these investigations are now shown in the new supplementary figure S5. The artificial outliers are injected by adding or subtracting a random value to the observed splicing metrics. We tested the injection with different values of the amplitude of these random deviations (0.2, 0.3, 0.5, and 0.7). Typically, the value of the dimension q giving the highest area under the precision-recall curve depended on the amplitude of the deviations, with higher dimensions (i.e. more complex models) performing better for the milder deviations. To not depend much on the value of these amplitudes, we therefore opted for drawing randomly the deviation amplitudes between 0.2 and the maximal possible amplitude a metrics can take (i.e. in order to reach 0 or 1). This typically led to an intermediate value of the optimal dimension q with near-optimal area under the precision-recall curve for each individual tested amplitude (Supplementary Fig S5). Moreover, we injected artificial outliers with deviations from the observed value itself or from its mean across samples. This did not influence substantially the results. We chose to inject deviations from the observed value.

- **Eq 19. The sum is integrating under the discrete probability distribution, right? If so, I missed why the $\frac{1}{2}$ is needed? Shouldn't the $\text{sum}_k(\text{BB})$ part be in the range $[0,1]$, so that either $\text{sum}_k(\text{BB}) < 1/2$ or $1 - \text{sum}_k(\text{BB}) < 1/2$?**

Response:

Thank you for spotting this. We made a mistake here in the formula. There is a -1 missing in the sum of the last term which we corrected. It should be $1 - \sum_{k=0}^{n(\text{bb})} [k-1]$ as we have to include the density of BB at k for both sides in a discrete distribution and with this it can happen that both sides are higher than $\frac{1}{2}$.

Additional data

- Due to independent discussions with other colleagues, we added a power analysis to this manuscript. This analysis shows that 30 samples are enough to detect most of the known pathogenic splicing outliers. This information is helpful for the design of experiments and in general for the diagnostic power. We added a paragraph at the end of the application to rare disease diagnosis section and the new Supplementary Figure S19.

Reviewer #1 (Remarks to the Author):

I've read the reviewer response and looked over the new additions. My concerns have been addressed and the new results are quite convincing in terms of FRASER's added value. I recommend acceptance.

Reviewer #2 (Remarks to the Author):

The authors were responsive to my concerns and in my view the manuscript benefitted from the additional analyses. The revised version provides good evidence of novel abilities for splice variation detection in short-read RNA-seq and I have no further comments.

Reviewer #3 (Remarks to the Author):

As this is the second round of review and I am a new added reviewer, my comments will be brief. I think reviewer 2's concerns were adequately addressed (which I was asked to evaluate). The authors did careful assessments of the performance of FRASER, and showed that their method outperformed the others. I think overall FRASER will be a useful tool for many in the community. I just have one comment.

In line 316, the authors "computed the enrichment at the gene level". I think it would be useful to also show the enrichment at the specific splicing event associated with the splice region or MMSplice predictions. Although I understand gene level enrichment does provide relevant information, splicing event specific enrichment is very interesting and has more relevant biological indications. If the enrichment at the specific splicing events is not high, it would be an alarming sign whether results from FRASER (or the other methods) are directly interpretable.

REVIEWERS' COMMENTS

Reviewer #1 (Remarks to the Author):

I've read the reviewer's response and looked over the new additions. My concerns have been addressed and the new results are quite convincing in terms of FRASER's added value. I recommend acceptance.

Reviewer #2 (Remarks to the Author):

The authors were responsive to my concerns and in my view the manuscript benefited from the additional analyses. The revised version provides good evidence of novel abilities for splice variation detection in short-read RNA-seq and I have no further comments.

Reviewer #3 (Remarks to the Author):

As this is the second round of review and I am a new added reviewer, my comments will be brief. I think reviewer 2's concerns were adequately addressed (which I was asked to evaluate). The authors did careful assessments of the performance of FRASER, and showed that their method outperformed the others. I think overall FRASER will be a useful tool for many in the community. I just have one comment.

In line 316, the authors "computed the enrichment at the gene level". I think it would be useful to also show the enrichment at the specific splicing event associated with the splice region or MMSplice predictions. Although I understand gene level enrichment does provide relevant information, splicing event specific enrichment is very interesting and has more relevant biological indications. If the enrichment at the specific splicing events is not high, it would be an alarming sign whether results from FRASER (or the other methods) are directly interpretable.

Response:

We thank reviewer #3 to have accepted to review the manuscript in the second round of revision. In response to this concern, we would like to underscore first that the gene-level enrichment was performed because variants show sometimes strongest effects on splice sites that are remote, for instance, due to competition between splice sites. Moreover, some methods only report aberrant splicing at the locus level rather than at the splice site level. We nonetheless agree that splice-site level enrichments are relevant and expected to hold. We now report splice-site-level enrichments for rare variants with a large MMSplice score (new Supplementary Figure S16). These enrichments are even more pronounced than at the gene level and remain consistently higher for FRASER than alternative implementations we have investigated.